# The Use of Edible Films Based on Sodium Alginate in Meat Product Packaging: An Eco-Friendly Alternative to Conventional Plastic Materials

**Roxana Gheorghita (Puscaselu)** [1,*], **Gheorghe Gutt** [2] and **Sonia Amariei** [2]

[1]  Department of Health and Human Development, Stefan cel Mare University of Suceava, 720229 Suceava, Romania

[2]  Faculty of Food Engineering, Stefan cel Mare University of Suceava, 720229 Suceava, Romania; g.gutt@usv.ro (G.G.); sonia@usm.ro (S.A.)

\*  Correspondence: roxana.puscaselu@usm.ro

**Abstract:** The amount of plastics used globally today exceeds a million tonnes annually, with an alarming annual growth. The final result is that plastic packaging is thrown into the environment, and the problem of waste is increasing every year. A real alternative is the use bio-based polymer packaging materials. Research carried out in the laboratory context and products tested at the industrial level have confirmed the success of replacing plastic-based packaging with new, edible or completely biodegradable foils. Of the polysaccharides used to obtain edible materials, sodium alginate has the ability to form films with certain specific properties: resistance, gloss, flexibility, water solubility, low permeability to $O_2$ and vapors, and tasteless or odorless. Initially used as coatings for perishable or cut fresh fruits and vegetables, these sodium alginate materials can be applied to a wide range of foods, especially in the meat industry. Used to cover meat products, sodium alginate films prevent mass loss and degradation of color and texture. The addition of essential oils prevents microbial contamination with *Escherichia coli*, *Salmonella enterica*, *Listeria monocytogenes*, or *Botrytis cinerea*. The obtained results promote the substitution of plastic packaging with natural materials based on biopolymers and, implicitly, of sodium alginate, with or without other natural additions. These natural materials have become the packaging of the future.

**Keywords:** pollution; zero-waste alternatives; sodium alginate; meat packaging

## 1. Introduction to the World of Plastics: The Need for Alternative Materials

The need to replace conventional plastic materials has become a subject of maximum interest, taking into account their intense polluting character and non-renewable nature. In modern society, these materials are found everywhere. Their excessive use is mainly due to their main characteristics: they are versatile, easy to process and manipulate, biologically inert, and can be obtained at low costs [1]. All these properties have promoted plastics intensely for various applications, from smartphones to the food industry or for 3D printing [2]. Nowadays, pollution due to plastics is found everywhere, including soil, oceans, drinking water, in human and animal bodies, and in the air. The production of plastics is expected to double over the next 20 years, surpassing to an alarming degree the current waste management and recycling capabilities [3]. It seems that plastic pollution has become the biggest environmental challenge of our time. Awareness of this problem has been achieved, with the help of non-governmental organizations and civil society, through numerous programs that publicized the current problem and helped to visualize the situation and to identify corrective measures. Even the European Union, in 2018, launched programs intended to develop strategies to reduce the use of plastics in order to protect the environment. There are two key problems that need to be solved:

(i) the financial dimension—only 5% of the value of the plastic materials is maintained in the economy, the rest being lost after the first use, which results in annual losses of 70–105 million euros, and (ii) their reduced degradation and overly long period of time that this takes—non-recycled plastic takes years to decompose, unlike other materials such as glass, paper, or metals. Researchers studying these topics have confirmed that, to date, traces of the first plastic materials discarded in nature can still found. Thus, millions of tons of non-degradable materials reach the environment. One of the most alarming outcomes is the massive pollution of the oceans (5–13 million tons/year) [4,5]. It is also estimated that obtaining and incinerating plastics produces about 400 million tonnes of $CO_2$/year. The strategy being adopted by European leaders is based on the idea of bioeconomy, the ultimate goal being to protect the planet and to defend the citizens. According to these plans, by 2030, all packaging on the European Union market will be recyclable and the consumption of plastics will be reduced. The organization is committed to reducing plastic waste, stopping mass storage, and investing in stimulating and innovating new materials [6]. Rethinking and improving this system requires cooperation and efforts from all actors involved, from plastic manufacturers to recyclers, retailers, and consumers. Without the active involvement of all key players, the ultimate goal cannot be achieved—a new plastic economy. The design and production of plastic must respect the ideals of reuse and recycling, and it must have the purpose of obtaining and promoting more sustainable materials. This action will reduce pollution due to plastics and its adverse impacts on people and the environment [7].

The food industry, along with the other industries, has become a major pioneer in terms of research directions towards substituting, partially or totally, the use of conventional materials. The FAO (Food and Agricultural Organization) reports that approximately 1.3 billion tonnes of food is wasted annually globally [8]. It should be noted that these polluting sources include household, commercial, industrial, and agricultural residues. Usually, remains of the food product they contained can be found on food packaging items, along with other biological substances. In this case, their recycling is impractical and economically inconvenient, the result being that tons of plastic packaging are thrown into the environment. A significant contribution to the increase of these figures is the excessive use of disposable packaging. Obtained from non-biodegradable and non-renewable materials, these are disadvantageous from all points of view, being considered a major source of pollution. The main role of food packaging is to protect products from physical, chemical, or biological factors, thus preventing damage. The maintenance of quality and safety, as well as extending shelf life, ensures product integrity throughout the food chain, from purchase to consumption. Depending on the proximity and contact with the food, such packaging can contain three layers: the first layer is in direct contact with the food and must provide protection during storage and distribution. The second layer is not in direct contact with food and protects it from physical changes (for example, plastic boxes or flexible bags). The third layer, such as pallets or adhesive films, usually contains films incorporated into the material. This layer provides additional protection against mechanical damage and environmental or air conditions. The proposed edible packaging generally fall into the category of the first-layer films [9].

## 2. Films and Coatings Used in the Food Industry

A real and viable solution for the eradication of this worrying phenomenon—pollution due to conventional single-use materials—is the development of completely biodegradable, even edible, materials obtained from bio-based polymers. Bio-based polymers are defined as polymeric materials produced from renewable raw materials (which may or may not be biodegradable). In addition to other features, special properties such as biodegradability, regenerability, relatively low-cost development, abundance in nature, non-toxicity, and biocompatibility have made them intensively used to obtain new innovative materials [10–12]. By 2011, the production of bio-based polymers had reached the threshold of 3.2 billion tonnes, and it is expected that after 2020 it will reach the threshold of 12 billion tonnes [13]. Bio-based polymers used in the food industry can be obtained from biomass, synthesized from bio-derivative monomers, or produced from microorganisms. In the choice of such polymers, the function that the film/coating has to fulfil is taken into account.

Moreover, the use of these films has generated increased consumer interest over minimally processed products [14,15]. Although it seems like a new method, it is part of a tradition; since the 16th century, lipid-based coatings have been used to cover pieces of meat [16]. The most common method of applying edible films and coatings is the use of an emulsion obtained from oil and waxes in water, which is sprayed directly on the surface of fruit in order to improve aspects of its appearance, such as gloss, color, and fineness, but also to control browning and loss of tissue water. A number of polysaccharide-based coatings, including alginates, carrageenan, cellulose derivatives, pectin, and starch derivatives have been used to improve meat quality during storage [17].

When a packing material (film, foil, thin layer, or coating) is an integral part of the food and is consumed with it, it is called an edible packaging. The coatings are applied to the surface of the food, while the films are independent structures that cover the food, being found on the surface or as a thin layer inside it. The edible nature of these materials can be achieved only when all the ingredients used are accepted by the food industry (food-grade ingredients), and the methods of obtaining the ingredients and the equipment used meet the requirements of food processing [18].

It is possible to retain product quality and freshness over the time required for commercialization and consumption if the materials and packaging technologies are correctly selected [19].

Although edible films and coatings could be classified as foods, this definition is not correct because they are not a finished product; they are not food products, they cannot be classified in a food class, and they do not have calculated nutritional value. Because they are both packaging and food components, edible films and coatings must have some characteristics, such as (a) a high sensory quality, (b) ensured mechanical efficiency and barrier property, (c) reduced permeability of water vapor, (d) biochemical, physio-chemical and microbiological stability [20], (e) non-toxic and not endangering consumers' health, (f) able to be obtained by simple technology using development processes that can be carried out at low costs [21], (g) non-polluting, (h) using cheap plant materials, (i) water resistance, (j) not producing excess $CO_2$, (k) easily emulsified and non-sticky, (l) not interfering with the quality of the food, (m) no taste or smell that can be detected at the time of consumption (if it has specific sensory characteristics, they must be compatible with the food), (n) reduced viscosity, (o) slightly transparent, but not like glass, and (p) capable of tolerating reduced pressures [22,23]. Although many functions of edible food packaging are similar to those of plastics (especially barrier properties against vapors and solutions), their use requires extra packaging, important for handling and hygiene reasons. The efficiency of biopolymeric films and coatings can be increased by various additions of natural substances with a role in improving the materials' physicochemical, mechanical, and microbiological characteristics [24]. This trend appeared mainly due to consumer demand for natural products. For this purpose, antioxidants, dyes, flavorings, essential oils, etc. have been added [25,26]. Research has also focused on intelligent packaging nano-materials with characteristics far superior to conventional films [19].

## 3. Sodium Alginate—a Basic Component of Bio-Based Polymer Materials

Bio-based polymer materials were invented in response to the need to replace conventional materials based on oil. They were made not to interact with the biological systems they contain. Of the majority of the hydrocolloids used, alginates have a special place, being one of the most popular and studied polysaccharides [27].

The FDA (US Food and Drug Administration) recognizes alginates as GRAS (generally recognized as safe) classified substances [28], and the EFSA (European Food Safety Authority) has authorized the use of alginate and related salts in *quantum statis* doses [6].

Alginates are isolated from the cell walls of brown algae (*Laminaria digtata/Ascophyllum nodosum*), where they are found in the form of sodium, calcium, and magnesium salts of alginic acid [29]; they can also be synthesized by microorganisms [30]. Alginate is a linear, anionic, water-soluble polysaccharide, and its most useful property—the formation of strong gels or polymers with low solubility (Figure 1)—is due to its ability to react with polyvalent metal cations, especially with calcium ions. This characteristic

has led to the improvement of mechanical properties, barrier properties, cohesiveness, and rigidity. By increasing the concentration of cations during the gelation of alginates, a dense structure with reduced porosity is formed and a decrease in the water content or the gel permeability appears [31]. Alginate solutions can form gels by pH reduction.

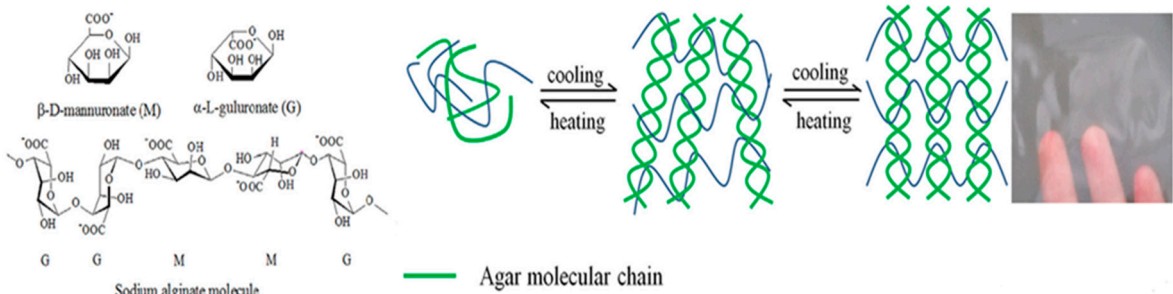

**Figure 1.** The sodium alginate gelling mechanism [32] (adapted from by Hou et al., Copyright 2019 Elsevier).

Due to their composition, alginates can form strong films with fibrous structure (in solid state), and are considered good filmogenic materials with characteristics comparable to those of conventional materials (Figure 2).

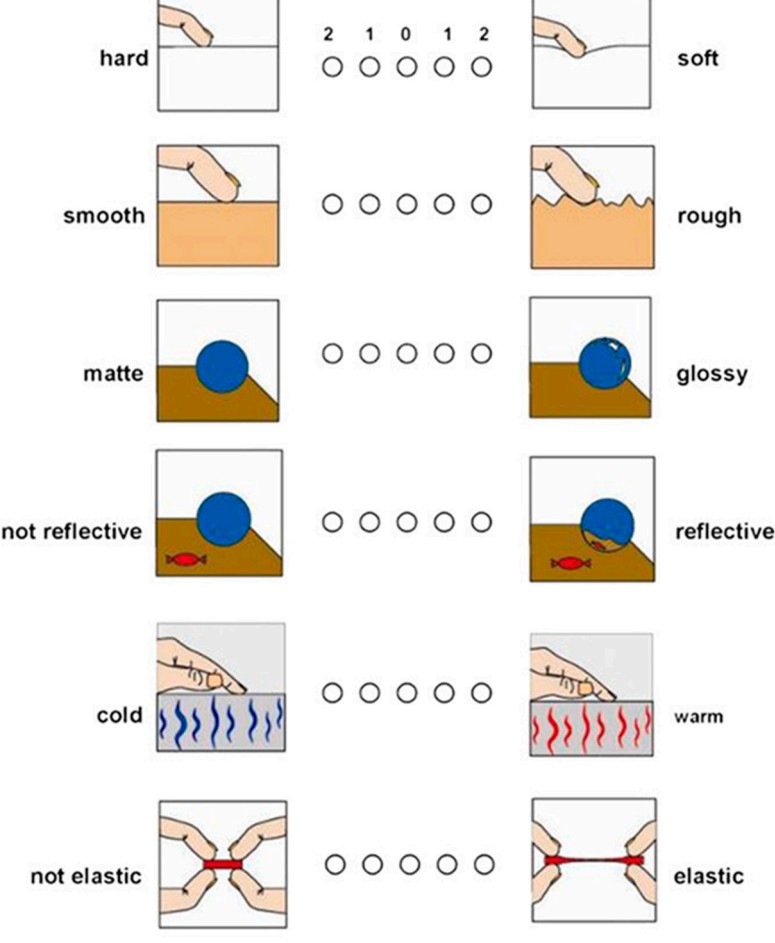

**Figure 2.** *Cont.*

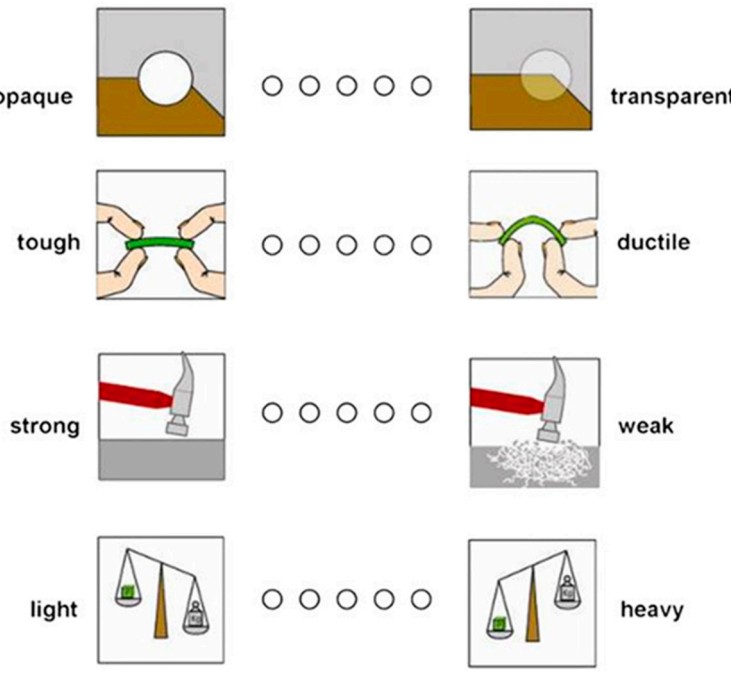

**Figure 2.** Material characteristics [33]. (Copyright 2012 Elsevier).

Sodium alginate is a nontoxic, biodegradable, biocompatible, and cheap hydrocolloid [34]. It may have and has many applications in the food industry as a packaging material for portioned products (Figure 3), limiting the dehydration of meat, as a thickening agent, in gel formation and as a colloidal stabilizing agent in the beverage industry, in the textile, pharmaceutical, and paper industries, and being used to obtain polymeric matrices used in the encapsulation of drugs, proteins, cells, and DNA.

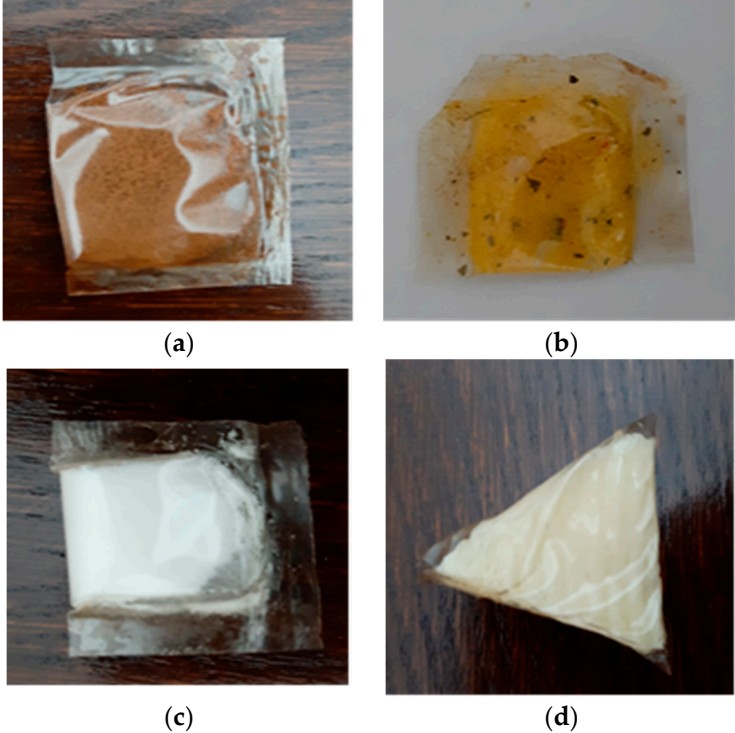

**Figure 3.** *Cont.*

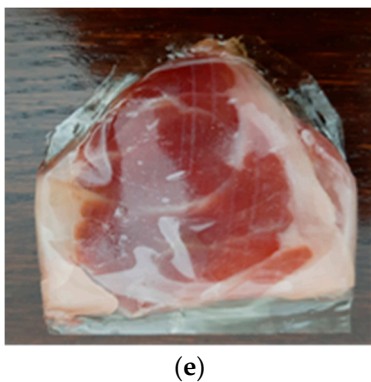

(**e**)

**Figure 3.** Applications of sodium-alginate-based edible films incorporated with *Stevia rebaudiana* for (**a**) soluble coffee, (**b**) dehydrated vegetables, (**c**) medicines in powder form, (**d**) cheese slices, and (**e**) meat slices—the pictured packaging film (both welded parts and excedentary parts) are entirely based on sodium alginate agar, and were plasticized with glycerol [35].

Sodium alginate is used in pharmaceuticals and medicine for encapsulation or as part of prolonged-release drugs [36–38], but also for gypsum, dental crowns, and prostheses, especially due to its hydrophilicity, pleasant taste, lack of odor, low cost, and ease of combination with other components and solubilization in the presence of saliva [39]. Alginate gels are currently used for cell transplants or for obtaining new tissues. They are used as substitutes for the organs or tissues of patients who have suffered certain losses [40,41]. They are also used in the textile industry to obtain bandages with special characteristics—bacteriostatic, antiviral, fungistatic, non-toxic, intensely absorbent, non-allergic, hemostatic, biocompatible, and allowing the respiration of the tissues and the incorporation of drugs, presenting superior mechanical properties necessary in this field [42].

Of all alginate types, sodium alginate can form films with certain specific properties: resistance, gloss, tastelessness or odorlessness, flexibility, water solubility, and low permeability to $O_2$ or oils (Figure 2).

From a structural point of view, sodium alginate is water-soluble, linear anionic polysaccharide consisting of monomeric units of 1–4-linked β-D-mannuronate (M) and α-L-guluronate (G). Sodium alginate has lots of hydroxyl groups, and less carboxylic acid groups. This provides the opportunity for the formation of intermolecular hydrogen bonding. Their ideal gelation mechanism is shown in Figure 2 [32].

In combination with glycerol, sodium alginate has been used to create fruit and vegetable coatings [43], thus contributing to the delay of degenerative processes and microbiological damage [44–46] by maintaining color, preserving the content of polyphenols and anthocyanins, and completely improving the quality of fruits after harvest [47–49]. Along with other hydrocolloids, it has been used to form coatings used in numerous subsectors of the food industry [31]. The easiest way to improve the characteristics of new materials is to mix in two or more biopolymers. Numerous studies have highlighted the synergistic character of sodium alginate with other biopolymers [50–53] (Table 1).

**Table 1.** Synergic characteristics of composite films obtained from polysaccharides.

| | AGAR | STARCH | CHITOSAN | SODIUM ALGINATE | CARRAGEENAN |
|---|---|---|---|---|---|
| **AGAR** | | +<br>low costs<br>-<br>brittle, rigid, gloss-free films, with high solubility, low tensile strength and elasticity | +<br>decreases color intensity, increases transparency and gloss; antibacterial properties | -<br>low tensile strength | -<br>increases hydration capacity and vapor permeability |
| **STARCH** | +<br>homogeneous microstructure, good barrier and mechanical properties | | +<br>increases mechanical strength, decreases vapor permeability and solubility; antibacterial effect | +<br>high water-holding capacity | +<br>improved mechanical properties<br>-<br>increases oxygen permeability |
| **CHITOSAN** | +<br>improved mechanical properties | -<br>reduced mechanical properties; low gloss and transparency | | +<br>allows compound encapsulation<br>-<br>low mechanical performance | +<br>allows the encapsulation of controlled-release compounds<br>-<br>increases hydration capacity and solubility |
| **SODIUM ALGINATE** | +<br>homogeneous, flexible, and fine films | -<br>high solubility, more brittle films | +<br>allows encapsulation of compounds | | +<br>vapor permeability decreases<br>-<br>low mechanical properties |
| **CARRAGE-ENAN** | +<br>reduces hydration capacity; more flexible and elastic films<br>-<br>reduces transparency | +<br>low cost<br>-<br>low flexibility and elasticity, high solubility | +<br>encapsulation of compounds | -<br>low elongation at break | |

## 4. Sodium Alginate—a Basic Component of Packaging Materials in the Meat Industry

The meat industry has always been one of the most polluting food industries. One intense environmental polluting factor is the packaging materials used. Plastics are especially preferred by manufacturers because of their properties. When a single applied layer cannot meet all the required characteristics, several laminate films are usually used. Besides the polluting nature of the materials when used independently, these complex films are extremely harmful due to the use of different compositions and the adhesives used for the overlap; all these practices make them impossible to sort and recycle once they become waste.

Edible films and coatings may be applied to meat and poultry products by foaming, dipping, spraying, casting, brushing, individual wrapping, rolling, or vacuum impregnation (Figure 4).

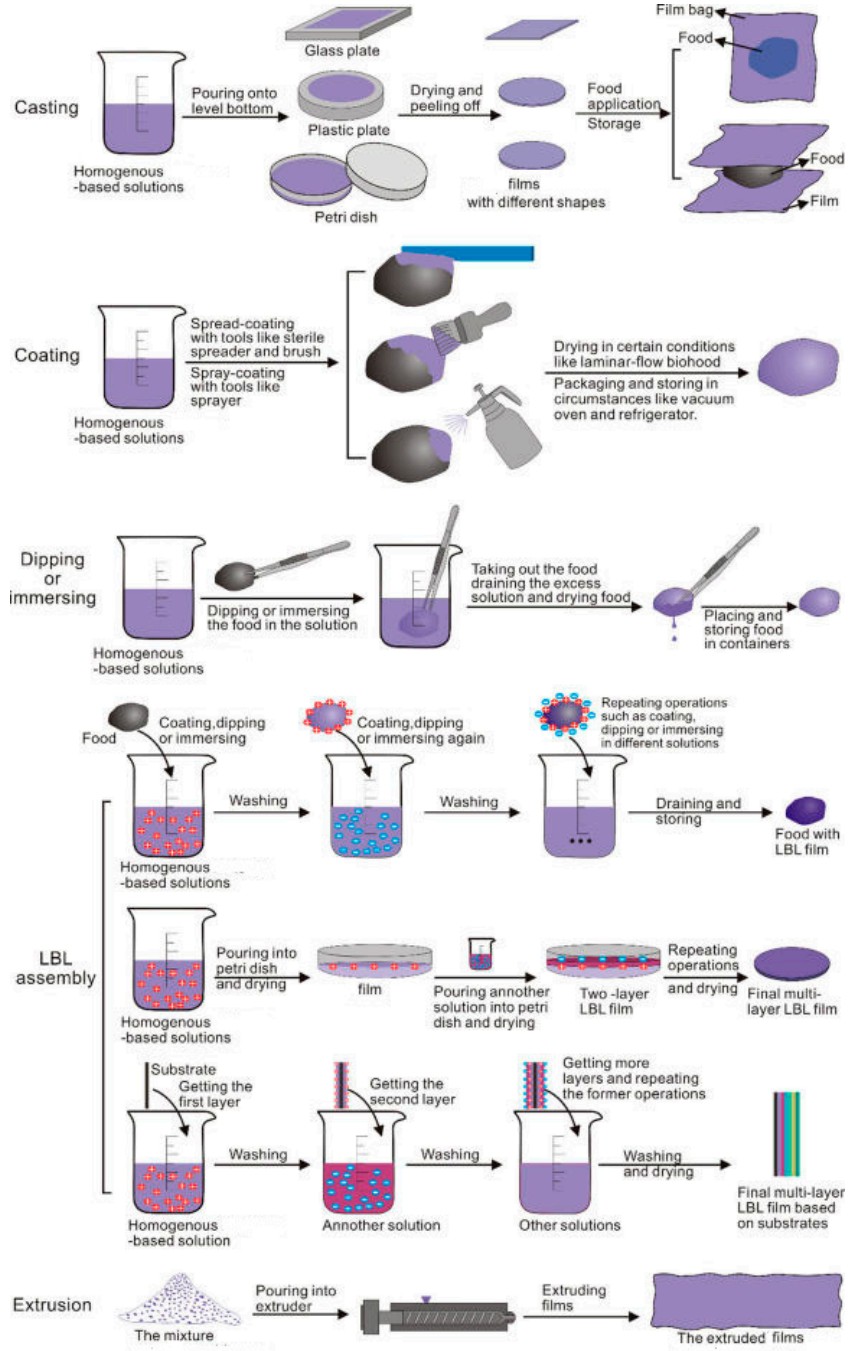

**Figure 4.** Ways of obtaining and applying films and coatings (adaptation by Wang, et al., 2018, [54]).

Foam application is used for those film-forming solutions that come in the form of an emulsion and requires the existence of a foaming agent; the coating is obtained with the help of compressed air; brushing is used for full application of the coating on the product surface; and a spray is generally used when the films are thin or when the coating is applied only on one side of the product [55]. The vacuum-impregnation method is used to enrich a product with vitamins or minerals [56]. Films are applied by the method of casting or extrusion when looking at composite or multilayer films [57]. In the meat industry, the packaging is first and foremost intended to protect the product from permeability to vapors and gases [58]. This should prevent mass loss and degradation of meat color [59]. Studies have shown the possibility of replacing these conventional materials with edible films based on biopolymers, especially polysaccharides, because they can extend the term of validity of meat and meat products by preventing dehydration, rancidity or browning of the muscle tissue [60]. Applied to meat products subjected to smoking or steaming, edible films dissolve on the surface of products, improving the structure and texture, and preventing moisture transfer [61]. In addition to its non-polluting, biodegradable, renewable nature, edible packaging may include various natural substances that play a role in maintaining and improving the quality of the packaged products, such as organic acids, essential oils, or plant extracts (Table 2), bacteriocins, antioxidants, colorings, flavorings, vitamins etc. The addition of antimicrobial substances to a food formula or in the materials used for food packaging is regulated by the FDA, which specifies the exact quantity permitted [62].

**Table 2.** Films based on sodium alginate with essential oil additives used in the food industry.

| Food | Essential Oils Added | Results | References |
|------|---------------------|---------|------------|
| Cheese | Pimpinella saxifrage | The addition of PSEO (1%–3%) in sodium alginate coating (2% sodium alginate+15% glycerol) was effective in reducing the weight loss, preserving pH and color and improving the oxidative and bacterial stability of the coated cheese. | [63] |
| Apples and pears | Cinnamon | Inhibition of growth and toxin production of *Aspergillus carbonarius* growth and ochratoxin A production. | [64] |
| Bighead carp fillets | Horsemint (*Mentha longifolia*) | Samples treated showed significantly lower lipid oxidation during the storage period and reduced degree of microbial deterioration. Antioxidant and antibacterial effects of sodium alginate coating and horsemint were more pronounced when a horsemint was used at 1% concentration. | [28] |
| Fresh-cut papaya | Thyme (*Thymus vulgaris*) and oregano (*Origanum vulgare*) | Retarded the degradation rate of physicochemical properties, improved microbiological food safety and had the highest sensory evaluation scores for fresh-cut papaya stored for 12 days at 4 °C. | [27] |
| Fresh-cut apple | Lemongrass | Inactivation of *Escherichia coli*. | [45] |
| Strawberry | Carvacrol | Carvacrol was effective against both *Escherichia coli* and *Botrytis cinerea*. | [65] |
| Fresh-cut apples | Cinnamon, clove, and lemongrass | The coatings applied on apple pieces maintained the physicochemical characteristics of the apple pieces for more than 30 days, decreased the respiration rate, reduced the *Escherichia coli* population by about 1.23 log CFU/g at day 0, and extended the microbiological shelf life by at least 30 days. | [66] |
| Strawberry | Citral and eugenol | Alginate edible with essential oils added improved the coatings in most cases, better preserved sensory and nutritional attributes, and reduced microbial spoilage. | [67] |
| Fresh-cut cantaloupes | Cinnamon bark oil and soybean oil | Cocktails of *Salmonella enterica*, *Escherichia coli* O157:H7, or *Listeria monocytogenes* inoculated onto cantaloupes were reduced to the detection limit and completely inhibited during 15 day storage with the coating treatment. | [68] |
| Fresh-cut pineapple | Lemongrass | The results indicate that an alginate-based edible coating formulation incorporated with 0.3% (*w/v*) lemongrass has potential to extend the shelf life and maintain quality of fresh-cut pineapple. | [69] |
| Fresh-cut melon | Cinnamon, palmarosa, and lemongrass | Palmarosa oil incorporated at 0.3% into the coating was shown to be a promising preservation alternative for fresh-cut melon, since it had a good acceptation by panelists, maintained the fruit quality parameters, inhibited the native flora growth, and reduced *Salmonella enteritidis* population. | [70] |

Antimicrobial packaging is a necessary and very important tool in protecting meat from contamination with pathogenic microorganisms. Contamination of meat products usually starts from the outer surface due to the microbial load resulting from improper processing and handling of the raw material and packaging. Antimicrobial films can prevent the contamination of meat and meat products during cold storage, but can also be used to inhibit microorganisms on the surface of freshly processed products, thus avoiding cross-contamination and increasing shelf life (Table 3). The prolonged, gradual release of antimicrobial substances from the structure of edible foils used for packing meat and meat products is a much more beneficial solution than their incorporation into the product [71,72].

The materials used for packing meat and meat products can be smart packaging, a principle based on two concepts: active and intelligent packaging. Active packaging refers to the existence of additives on the surface of a packaging material, attached inside it or incorporated into its structure, which play a role in maintaining the quality and extending the validity term of a product. Smart packaging systems, however, are designed to monitor certain characteristics of either packaged products or the environment in which they are found [73], and to provide information on product quality and consumption safety [74]. They can be sensors (electronic noses, biosensors, or chemical sensors) or indicators (freshness, time–temperature, integrity) [13,75].

**Table 3.** Alginate-based films and coatings with applicability in the meat industry.

| Food | Film | Results | References |
|---|---|---|---|
| Chicken breast meat | Alginate-based film with black cumin | Indicated antimicrobial activity against *Escherichia coli*, less variation in pH, lower color changes for chicken breast meats over 5 days of storage at 4 °C. | [76] |
| Pastirma | Sodium-alginate- and gelatin-based coatings | The color of pastirma was preserved after coating with gelatin or alginate, which will make the dry cured meat more attractive to consumers. | [77] |
| Sausages | Sodium alginate | *Enterobacteriaceae* microorganisms were inhibited by about 2 log $CFU/cm^2$ on sausage samples covered with experimental sodium alginate coatings. | [78] |
| Raw silver carp (*Hypophthalmichthys molitrix*) fillets | Sodium alginate coating containing *Mentha spicata* essential oil and cellulose nanoparticles | Retarded the growth of mesophilic and psychrotrophic bacterial population for up to 7 days and enhanced the shelf life of minced beef meat compared to the control; the shelf life of silver carp fillets using coatings was significantly improved for up to 14 days compared to the control group. | [79] |
| Refrigerated trout (*Oncorhynchus mykiss*) fillets | Sodium-alginate-based films with resveratrol | Enhanced the shelf life and exhibited strong antioxidant activities at very low concentrations; it also led to stable and unchanged sensory properties such as odor, color, flavor, and general acceptability. | [80] |
| Fish fillet | Alginate-based nanocomposite films with marjoram, clove, cinnamon, coriander, caraway, and cumin | Inhibited the growth of *Listeria monocytogenes* in the following order: marjoram > clove > cinnamon > coriander > caraway > cumin essential oils. | [81] |
| Beef steaks | Alginate-based films with rosemary and oregano essential oils | The edible coatings decreased lipid oxidation of the meat compared to the control; significantly decreased color losses, water losses, and shear force compared to the control; and had a significant effect on consumer perception of odor, flavor, and overall acceptance of the beef. | [82] |
| Carp fillet | Alginate/carboxyl methyl cellulose composite coating incorporated with clove essential oil | Maintained silver fillet shelf life up to 16 days without any significant loss of texture, odor, color, or overall acceptability, and had lower bacterial count, TVB-N, and lipid oxidation rate in comparison to control. | [83] |

**Table 3.** *Cont.*

| Food | Film | Results | References |
|---|---|---|---|
| Chicken meat | Sodium alginate coating incorporated with nisin, *Cinnamomum zeylanicum*, and rosemary essential oils | Longer storage of chicken breast at refrigerated temperatures (4 °C); reduced the degradation of bioactive compounds and maintained the nutritional quality. | [84] |
| Baked ham and bologna sliced | Alginate films and essential oils of oregano, cinnamon, savory | Reductions populations of *Salmonella enterica* and *Listeria monocytogenes* in bologna and baked ham slices when applying alginate films with cinnamon essential oil. | [85] |
| Fish fillet | Gelatin–alginate film containing oregano essential oil | Study showed that film was an effective antimicrobial suitable for potential food packaging applications; use of the OEO blend film delayed bacterial growth over 15 days of storage. | [86] |
| Beef muscle | Sodium alginate and oregano, cinnamon, savory oils | After 5 days of storage, films containing oregano or cinnamon essential oils were the most effective against *Salmonella typhimurium*. | [87] |
| Fish fillets | Alginate–calcium coating with cinnamon and nisin | Cinnamon in alginate–calcium coating treatments could efficiently maintain quality during storage, but colors of fish fillets were evidently changed due to the color of cinnamon. | [88] |
| Rainbow trout fillets | Alginate coating containing lactoperoxidase system and *Zataria multiflora Boiss* essential oil | Alginate coating, when used with no antimicrobial agent, had a supportive effect on the growth of *Listeria monocytogenes* and *Escherichia coli O157:H7* pathogenic bacteria. | [89] |

The packaging of a food product influences the consumer in their choice and willingness to pay for it.

Thus, there are two avenues by which to tempt a consumer: either by using attractive images on the front of the packaging, or by the existence of transparent areas in the packaging material so that the consumer can observe the food. The use of hydrocolloid films based on sodium alginate can facilitate this aspect, as the product can be evaluated over its whole surface. Research has shown that, lately, producers have been choosing the second option, because it seems that the product packed in such materials is not only preferred by consumers over others, but also encourages them to buy food in higher quantities [90]. These materials are easy to apply to ready-to-eat foods, which are extremely modern today. They can also be successfully used for wrapping packages containing a single portion, made precisely to avoid wasting food [91]. Only the use of edible or biodegradable materials can reduce the amount of conventional disposable plastic packaging used for this purpose.

## 5. Conclusions and Future Perspectives

In conclusion, one thing has become clear: regular plastic packaging materials used in the food industry, often for single uses, are polluting. Viable solutions to this problem include the use of alternative packaging materials—natural, obtained from bio-based polymers, biodegradable, and even edible. Studies and research in this regard highlight the use of films and coatings successfully applied to products across the entire food chain.

Sodium alginate has proven effective in the food industry, but also in other important human industries, such as medicine and pharmaceutics. Used as a packing material in the meat industry, it protects the product from the evaporation of water content, color loss, and gas transfer and prevents microbial contamination. For the continuous improvement of these characteristics, further research is required to identify new types of films and foils with improved features, which could maintain and even increase the quality of products throughout the food chain, aimed, in particular, at increasing shelf life. Similarly, it is possible to identify natural substances with nutritional, pre-/probiotic, antioxidant, or anti-browning agents that can ensure the quality of packaged products for a longer period.

Developing awareness campaigns for producers and consumers can have a positive effect in amplifying the use of new packaging materials.

This is the packaging of the future.

**Author Contributions:** Concept, method, investigation and draft writing—R.G.; Analysis—G.G. and S.A.; Writing editing—S.A.; Supervision—G.G. and S.A.; Project management—G.G. All authors have read and agreed to the published version of the manuscript.

**Funding:** This work was supported by contract no. 18PFE/16.10.2018 funded by Ministry of Research and Innovation within Program 1—Development of national research and development system, Subprogram 1.2—Institutional Performance -RDI excellence funding projects.

**Acknowledgments:** The authors are thankful to the "B&V The Agar Company", Italia, for their support to the development of this study and for providing the materials necessary for successfully conclude this research.

**Conflicts of Interest:** The authors declare no conflict of interest.

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
