# Peer review of "The Use of Edible Films Based on Sodium Alginate in Meat Product Packaging: An Eco-Friendly Alternative to Conventional Plastic Materials"

_coatings, doi:10.3390/coatings10020166_

Round 1

Reviewer 1 Report

The topic of the manuscript "The use of edible films based on sodium alginate in meat products packaging: an eco-friendly alternative to conventional plastic materials" presents both academic and industrial interest.

The selected references offer a correct image of the stat-of-the-art on this topic.

However, the present manuscript contains several:

important confusions misunderstandings and misapplication of basic terms describing the materials; confusing English sentences; information that are not at their place in such a review (for instance the large description of basic data form standards concerning the mechanical requirements for food packaging materials).

Some examples figure in the attached file.

For all these reasons, I suggest a major revision of the manuscript.

Author Response

Dear Reviewer,

I understood the comments that you made and I tried to modify the text as you suggested. I hope you will find the changes made in the new version of the manuscript.

As you have suggested, I deleted the Section 5, related to film testing methods. My initial idea was to gather in a single document all the procedures of films' testing methods, as I used them in my laboratory research. In this way, the researcher-reader has at hand all the methods and materials in order to be able to test the films and coatings. 

Thank You for your comments!

Reviewer 2 Report

I have read the present review manuscript entitled “The use of edible films based on sodium alginate in meat products packaging: an eco-friendly alternative to conventional plastic materials” by Gheorghita et al., and my comments are the following ones:

Abstract

Abstract is too general. It should include some relevant information regarding the use of edible packaging for meat.

Keywords are missing. Section 1

Related to Section 1, some statements need to be rewritten because they are too long making them difficult to understand (i.e. the final sentence in the paragraph 1 before the reference 8 “Without the active involvement of each key player, the correct purpose cannot be achieved – a new plastic economy, where the design and production of plastic respects the reuse and recycling, obtaining and promoting more sustainable materials, which will reduce the pollution due to plastic and its adverse impact on people and the environment”, lines 5-7 in paragraph 2 “Usually, on the packaging are found remains of the food product they contain, but also other biological substances, so that their recycling is impractical and economically inconvenient, the result being that tons of plastic packaging are thrown in environment, thus increasing the problem of waste every year”).

Section 2

In Section 2, I suggest to clearly define the differences between films and coatings.

Section 3

Related to Section 3, acronyms need to be defined (GRAS and EFSA). Images from Figure 2 need to be numbered and defined into the main text.

Section 4

In Section 4, figure 4 is of very poor quality.

Section 5

Regarding to Section 5, all the properties and techniques described are not specific for meat packaging. They are the general methods used for every kind of films and coatings. I highly recommend avoiding this section.

Instead, I suggest discussing results contained in Table 3 (they are just mentioned in the Table) due to this is a review regarding the use of edible sodium alginate packaging in meat and no discussion about this has been made in the entire manuscript. Properties mentioned in section 5, should be discussed according to the cited results in Table 3 instead of describing how to measure them in order to make this review more useful.

Conclusions need to be rewritten for better understanding. References need to be revised in order to the journal citation style. Adapt the manuscript to the journal format style.

Author Response

Dear Reviewer,

We made the changes that you suggested. 

So, we have introduced in the abstract section clear notions that refers to the advantages of using sodium alginate-based films in meat industry.

Similarly, we added keywords.

Section 1: we divided the phrases in order to be easier to read and understand.

Regarding section 2, I think the differences between films and coatings is noted in the following statement: " The coatings are applied on the surface on the food, while the films are independent structures that cover the food after it is obtained, being found on the surface or as a thin layer inside it.". Delimitations between films and coatings also occur in areas with examples of their use in the food industry.

Section 3 - we defined the terms; we numbered the images.

Section 4 -  quality improved.

Section 5 was deleted. 

Thank You for your review! 

Reviewer 3 Report

The subject of this review article is very interesting and usefull for the future of packaging. Unforunatelly the section 5 "Evaluation of the Performances of Films and Coatings in the Meat Industry" is purly writtened and must be fully revised.

I suggest the authors provide specific results from the literature for all or some of the properties are listed in section 5. Specific experimental result is the main perpose for someone to read a review paper and are more usefull than theoretical analysis of packaging performance properties.

Also the analysis of Figure 4 must be increased.

Author Response

Dear Reviewer,

Thank You for your comments regarding our manuscript.

We deleted section 5. Our idea was to make available to the researcher a unique document that would include all the methods that can be used for testing films and coatings.

We improved the Figure 4.

Thank you for your review!

Round 2

Reviewer 1 Report

The revised version of the manuscript is largely improved, however some misued of terms still subsist.

Please replace 'plastic' with "plastic materials" (PLURAL, as there is no ONE plastic but a large variety);

Please avoid ALL the confusions between biodegradable polymers, biopolymers, bio-based materials, biomaterials, considering the definitions already send. I have indicated with yellow some of the misused termes, to be replaced with the correct terms.

The same for "biofilm" notion, which has a very specific meaning that is not at all appropriated in the context of Legend (please see my comment on pdf version).

The manuscript is a good compilation of the existing data on alginates for food packaging. However, the misuse of specific scientific terms is penalizing the quality of the entire manuscript. Please do the corrections in accord with the official scientific definitions from Polymer Science.

I recommand this manuscript for publication, after systematic replacement of the wrong scientific terms.

Author Response

Dear Reviewer,

Thank You for all your comments. The attached document that you send me, it was really helpful. 

I hope this version of the manuscript complies with your requirements.

Reviewer 2 Report

I suggest this revised version of the manuscript for accepting in the present form but I recommend some modifications before publication: 

The manuscript needs to be in the format of the journal. It seems that the journal format is starting in Section 4 while should be starting from the beginning.  In Page 3, the caption for the figure has been missing and needs to be present.  In Page 5 it is appearing the second figure of the manuscript but it is called figure 7. 

Please, revise these items before publication in order to provide an organized format according to the journal style and guidelines. 

Author Response

Dear Reviewer, 

Thank You for the comments.

I hope everything is fine now. 

Reviewer 3 Report

You have respond in all reviewrs suggestions. I suggest accept in current form.

Author Response

Thank You!